# Analysis of Vibration Frequency and Direction for Facilitating Upper-Limb Muscle Activity

**DOI:** 10.3390/biology12010048

**Published:** 2022-12-27

**Authors:** Cheng-Hua Ni, Yueh-Hsun Lu, Li-Wei Chou, Shu-Fen Kuo, Chia-Huei Lin, Shang-Lin Chiang, Liang-Hsuan Lu, Xin-Miao Wang, Jia-Lan Chang, Chueh-Ho Lin

**Affiliations:** 1Department of Nursing, Center for Nursing and Healthcare Research in Clinical Practice Application, Wan Fang Hospital, School of Nursing, College of Nursing, Taipei Medical University, Taipei 11031, Taiwan, R.O.C.; 2Department of Radiology, Shuang-Ho Hospital, Taipei Medical University, New Taipei City 235041, Taiwan, R.O.C.; 3Department of Radiology, School of Medicine, College of Medicine, Taipei Medical University, Taipei 11031, Taiwan, R.O.C.; 4Department of Physical Therapy and Assistive Technology, National Yang Ming Chiao Tung University, Taipei 112304, Taiwan, R.O.C.; 5School of Nursing, College of Nursing, Taipei Medical University, Taipei 11031, Taiwan, R.O.C.; 6Department of Nursing, Tri-Service General Hospital, School of Nursing, National Defense Medical Center, Taipei 114202, Taiwan, R.O.C.; 7Department of Physical Medicine and Rehabilitation, Tri-Service General Hospital, School of Medicine, National Defense Medical Center, Taipei 114202, Taiwan, R.O.C.; 8Faculty of Humanities, Zhejiang Dong Fang Polytechnic College, Wenzhou 325011, China; 9Department of Physical Medicine and Rehabilitation, Shuang Ho Hospital, Taipei Medical University, Taipei 235041, Taiwan, R.O.C.; 10Master Program in Long-Term Care, College of Nursing, Taipei Medical University, Taipei 11031, Taiwan, R.O.C.; 11International Ph.D. Program in Gerontology and Long-Term Care, College of Nursing, Taipei Medical University, Taipei 11031, Taiwan, R.O.C.; 12Center for Nursing and Healthcare Research in Clinical Practice Application, Wan Fang Hospital, Taipei Medical University, Taipei 116079, Taiwan, R.O.C.

**Keywords:** vibration, direction, frequency, muscle, electromyography, upper limb

## Abstract

**Simple Summary:**

Health promotion is important in older adults. However, aging-induced frailty results in poor muscle activity in the upper limbs, leading to activity impairments. Fortunately, recent studies report that vibration is a safe approach for improving muscular function; however, different frequencies and directions of vibrations can result in inconsistencies in muscle function improvement, and further investigation is needed. We developed a handheld vibrator to determine the effect of vibration frequency and direction on upper-limb muscle activation. Nineteen qualified participants were exposed to vertical and horizontal vibrations with 0, 15, 30, 45, and 60 Hz frequencies for 60 s each. Surface electromyography measured the activities of the flexor digitorum superficialis (FDS), flexor carpi radialis (FCR), extensor carpi ulnaris (ECU), extensor carpi radialis (ECR), biceps, triceps, and deltoid anterior muscles. Muscle activity was significantly induced under vibration conditions in both vertical and horizontal directions. The 45-Hz horizontal vibration induced maximum muscle activations for the FDS, ECR, ECU, biceps, and triceps. The 60-Hz vertical and 30-Hz horizontal vibrations facilitated maximum muscle activations for the FCR and deltoid anterior, respectively. We therefore suggest different protocols of vibration for specific weak muscles to improve muscle function in the upper limbs of older adults.

**Abstract:**

We aimed to determine the effect of vibration frequency and direction on upper-limb muscle activation using a handheld vibrator. We recruited 19 healthy participants who were instructed to hold a handheld vibrator in their dominant hand and maintain the elbow at 90° flexion, while vertical and horizontal vibrations were applied with frequencies of 15, 30, 45, and 60 Hz for 60 s each. Surface electromyography (EMG) measured the activities of the flexor digitorum superficialis (FDS), flexor carpi radialis (FCR), extensor carpi ulnaris (ECU), extensor carpi radialis (ECR), biceps, triceps, and deltoid anterior muscles. EMG changes were evaluated as the difference in muscle activity between vibration and no-vibration (0 Hz) conditions. Muscle activity was induced under vibration conditions in both vertical and horizontal (*p* < 0.05) directions. At 45 Hz, FDS and FCR activities increased during horizontal vibrations, compared with those during vertical vibrations. ECU activity significantly increased under 15-Hz vertical vibrations compared with that during horizontal vibrations. Vibrations from the handheld vibrator significantly induced upper-limb muscle activity. The maximum muscle activations for FDS, ECR, ECU, biceps, and triceps were induced by 45-Hz horizontal vibration. The 60-Hz vertical and 30-Hz horizontal vibrations facilitated maximum muscle activations for the FCR and deltoid anterior, respectively.

## 1. Introduction

Muscle activation and related upper-limb functions play important roles in performing activities of daily living such as feeding, bathing, and dressing. Aging or neurologic disorders may result in upper-limb motor control and functional impairment [1]. Many studies have indicated that older adults with degenerative conditions exhibit significant motor function deficits, which lead to frailty and long-term disabilities [2,3,4]. However, studies have also revealed that the frailty could result from the physiologic age-dependent changes in vicious loops [5,6] or lifestyle (sedentary behaviors) [7], leading to poor muscle strength and muscle atrophy (sarcopenia) [5,6]. This could result in muscle weakness, low physical activity, and impaired upper-limb function [5,6,7,8,9,10,11,12,13] and could increase dependency levels in daily living for older persons. Moreover, older adults are reportedly predisposed to developing muscle atrophy and weakness. Greenlund et al. reported that mortality due to stroke—one of the most common neurologic diseases in older adults—has decreased [14]; however, stroke-induced motor function deficits result in paretic upper limbs [15,16], which alter the daily functioning and quality of life and cause long-term disabilities in patients with stroke [17,18]. Only 15% of patients in the acute and subacute stages of stroke who undergo traditional rehabilitation interventions recover their paretic upper-limb motor function [19]. An appropriate rehabilitation intervention is required to improve upper-limb neuromuscular function and motor recovery in older adults and people after stroke.

The vibration approach is reportedly effective, easy, satisfactory, and safe for improving muscle strength, power, and function [20,21,22,23]. The mechanism for muscle function improvement is based on facilitating the activation of the efferent Ia muscle fibers during vibration approaches, leading to α-motor neuron excitation, which produces more muscle force output [24,25]. The vibration approach included asking the participants to sit on a chair or kneel on the ground and to place their hands or elbows on the vibratory platform [22,26,27,28,29,30,31,32] or to directly sit on the vibratory platform with two hands in the sideways position on the vibratory platform [33]. Studies have shown that different vibration approach protocols (frequencies between 5 and 60 Hz) and vibration directions (horizontal or vertical) lead to improvements in muscle strength [25,34,35,36]. However, contradictory findings were also reported in recent studies [22,37]. Vibration transmission is a complex process that is influenced by biomechanics [27,38]; beside the vibration frequencies, the different postures during vibration approaches may also affect upper-limb muscle activation [22,27]. This indicates that the force translation to the upper limbs may change because different postures may result in different vibration directions from the vibrator. Based on previous findings regarding the mechanism and inconsistent findings of vibration approaches, we speculated that vibration frequency and directions constitute key factors that facilitate muscle activation during vibration approaches. In addition, the outcome of vibration application on the upper-limb musculature remains inconclusive because of differences in the vibration stimuli applied and limited appropriate equipment; only a few studies have applied vibration-induced muscle activations in the upper limbs [27,28,39,40,41]. Furthermore, many studies have suggested that the efficacy of appropriate equipment with optimal vibration protocols to induce muscle activation should be established before applying these protocols clinically [23,24,25,42,43,44,45]. Furthermore, recent studies have suggested that vibration-induced changes in neuromuscular activation during different vibration stimuli can be evaluated directly using the root mean squares of surface electromyographic (EMG) signals (EMGrms), which reveal the effect of vibration stimuli on muscle strength enhancement [27]. Therefore, this study aimed to develop a handheld vibrator to determine the effect of vibration frequency and direction on upper-limb muscle activation.

## 2. Material and Methods

### 2.1. Participants

The sample size requirement was calculated using G*power (version 3.1.9.2, Heinrich-Heine-Universität, Düsseldorf, Germany) using an effect size of 0.25, an alpha of 0.05, and a power of 0.80. A total sample size of 34 participants was required. However, due to the coronavirus disease 2019 (COVID-19) pandemic, only 19 adults were recruited. The inclusion criteria required that participants be able to follow the researcher’s instructions and study procedures, be healthy with no cognitive disorders that could affect the vibration approach performance, be able to steadily hold a handheld vibrator using the dominant hand without pain or discomfort, and have good cognitive function. We excluded patients with acute or chronic neurologic or orthopedic impairments and those who experienced discomfort or had undergone surgery in the upper limbs within 6 months prior to the study onset. The dominant hand for each participant was considered the hand that was used to sign the informed consent form.

This study was conducted in communities and approved by the institutional review board of Taipei Medical University (approval number: N202007048). All participants signed an informed consent form before participating in this study. This study was carried out in accordance with the Declaration of Helsinki.

### 2.2. Research Device and Data Processing

A handheld vibrator (measuring 37 × 8.3 × 10.5 cm; weight, 1 kg) was designed and customized by ACCU BALANCES CORP. (New Taipei City, Taiwan). The handheld vibrator motor (ZYT3424D110; 1.833 V = 1 Hz) produced vibrations with a 5-mm amplitude and 0–60-Hz frequency range, and its validity was confirmed by analyzing the vibration frequencies generated by the vibrator between 0–60-Hz with APDM-collected vibration frequency data. The vibration frequency could be controlled using a vibration controller (rotary potentiometer) located on the right side of the vibration frequency control box; a frequency mechanical pointer located on the left side of the box showed the voltage applied to the vibration motor, which is correlated with the frequency measurement via APDM (Figure 1). To determine the effects of vibration on the upper limbs, a red line target indicated the position for holding the vibrator during vertical (up and down vibration in the sagittal plane) and horizontal (medial and lateral vibration in the horizontal plane) vibration approaches (Figure 2). The handheld vibrator was applied to generate vibration force to the entire upper limb rather than a specific muscle during the vibration tests.

The research device includes a handheld vibrator and vibration frequency control box. The researcher controls the vibration frequency, which is displayed on the mechanical pointer, using the vibration controller.

Participants used their dominant hand to hold the vibrator at the red line. Participants held the vibrator in two different ways (Figure 2a,b) so that the direction of vibration was either up–down or lateral–medial.

### 2.3. Experimental Procedures and Positioning of Participants

All participants were instructed to sit back on a high-fixed, no-arm support chair with their feet positioned flat on the floor. The dominant shoulder was positioned sideways, slightly apart from the trunk, and the elbow was fixed at 90° flexion as the standard vibration position (Figure 3). Thereafter, the researcher instructed the participants to hold the red line on the vibrator firmly with the dominant hand. Participants randomly performed all vibration tests comprising five vibration frequencies (0, 15, 30, 45, and 60 Hz) in the vertical and horizontal vibration directions. Moreover, participants were instructed to perform all vibration approaches while maintaining the elbow in the standard vibration position.

Based on our review of previous clinical studies on improving muscle strength, we used vibration protocols with vibration exposure between 30 and 60 s and with resting intervals between 15 and 60 s [46,47,48,49]. Studies have reported that fatigue [50], muscle adaption, and injury risk [24] can increase when vibrations last more than 1 min. Previous research also reported that the vibration approach has 30 s of maintained effects on facilitating muscle excitation [51]. Therefore, the vibration approach was performed for 60 s with a 1-min resting interval between sessions to avoid muscle fatigue.

### 2.4. EMG Analysis

Seven muscle groups, including the flexor digitorum superficialis (FDS), flexor carpi radialis (FCR), extensor carpi radialis (ECR), extensor carpi ulnaris (ECU), biceps, triceps, and deltoid anterior, were selected to represent the performance of the upper limbs [22,52]. EMG signals were measured using the BTS FREEEMG 1000 with EMG-BTS EMG-Analyzer^®^ (BTS Bioengineering, Milan, Italy) during vibration exposure, at a sampling frequency of 1000 Hz. Based on the Surface Electromyography for the Non-invasive Assessment of Muscles guidelines, pairs of bipolar Ag-AgCl electrodes (H124SG Covidien, Minneapolis, MN, USA) were placed over the belly of each muscle, with an inter-electrode distance of 2 cm [53]. Before placing the electrodes, each participant’s skin was thoroughly cleaned with alcohol swabs [27]. Electromyography (EMG) signals were amplified with a gain of 1000. EMG post-processing was performed using the EMG-Analyzer. We used bandpass EMG signals (20–400 Hz); a notch filter was used to remove the noise from the power line (60 Hz). EMGrms, with a 100-ms window, was used to process EMG data for each muscle during five vibration approaches sessions [27,30,39,54]. Further, the EMGrms values were normalized with respect to the percentage of maximum voluntary contraction of each corresponding muscle for each participant [39].

### 2.5. Statistical Analysis

The Shapiro–Wilk test was performed to test the normality of the sample data. The Mann–Whitney U test was performed to confirm whether the vibration approaches had a significant impact on individual muscle activation in the vertical and/or horizontal vibration directions. Further, the Friedman test was performed to compare the increase in muscle activation between vibration (15, 30, 45, and 60 Hz) and no-vibration (0 Hz) conditions. Lastly, the Wilcoxon signed-rank test was performed to compare vibration direction-related changes in muscle activation for each muscle with different vibration frequencies in the vertical and horizontal vibration directions. The alpha level was set at 0.05. Statistical Package for the Social Sciences software (version 17.0, SPSS Inc., Chicago, IL, USA) was used for statistical analysis.

## 3. Results

### 3.1. Participants

Nineteen adults (age 38.2 ± 14.0 years; 14 female and 5 male) were recruited in this study (Table 1).

### 3.2. EMG Analysis

All variables were assumed to be non-normally distributed (all *p* < 0.001, respectively), including FDS, FCR, ECR, ECU, biceps, triceps, and deltoid anterior.

On comparing the vibration direction-related changes in muscle activation for upper-arm and shoulder muscles, the results showed no statistically significant differences in muscle activation for the biceps, triceps, and deltoid anterior muscles at 0, 15, 30, 45, and 60 Hz vibration frequencies in the horizontal and vertical vibration directions (Table 2; Figure 4).

However, the FDS and FCR muscles were more activated during the 45-Hz vibration approach in the horizontal vibration direction than in the vertical direction (*p* < 0.05). In addition, the ECU muscles were more activated during the 15-Hz vibration approach in the vertical vibration direction than in the horizontal direction (Table 2; Figure 5).

The vibration approaches significantly impacted all individual muscle activations in the vertical (*p* < 0.05) and horizontal (*p* < 0.05) vibration directions (Table 2).

All muscle groups had significantly facilitated muscle activation in the vibration condition compared with that in the no-vibration condition, in both vibration directions (Table 2). Furthermore, compared with no-vibration (0 Hz), for both vertical and horizontal vibration directions, the maximum muscle activation for FCR was facilitated by 60 Hz vertical vibration (*p* < 0.001); the maximum muscle activations for FDS, ECR, ECU, biceps, and triceps were induced during 45 Hz horizontal vibration (*p* < 0.001), respectively. For the deltoid anterior, the maximum muscle activation frequency was induced during 30 Hz horizontal vibration (*p* < 0.001) (Table 2).

## 4. Discussion

Many studies have used whole-body vibration devices to investigate the impact of vibration approaches on upper-limb muscle activation [22,26,27,28,29,31,32]. In contrast, this study developed a frequency-controlled handheld vibrator with a focus on upper-limb muscle activation. We found that vibration approaches with specific frequencies (15, 30, 45, and 60 Hz) had a positive effect on muscle activation in all seven muscle groups i.e., FDS, FCR, ECR, ECU, biceps, triceps, and deltoid anterior in both the vertical and horizontal directions, unlike in the absence of vibration (0 Hz). Furthermore, horizontal and vertical vibrations had a significant facilitatory effect on the activation of upper-limb flexors (FDS and FCR muscles) and extensors (ECU muscles), respectively. To the best of our knowledge, this is the first study to report that different vibration directions induce the activation of different upper-limb muscles.

### 4.1. Vibrator Types and Applications in Upper-Limb Muscle Activity Induction

Most previous studies used vibratory platforms as the vibrator, and participants were instructed to take specific positions for upper-limb muscle activation. For instance, participants were instructed to sit on a chair or kneel on the ground and then support their upper limbs over the vibratory platform with their hands or elbows shoulder-width apart [22,26,27,28,29,30,31,32]. In other studies, participants were instructed to hold the handrail on the top of the vibratory platform [27] or sit on the vibratory platform with two hands in the sideways position [33]. The approaches used in previous studies may or may not have induced upper-limb muscle activation; however, they could cause vertigo or discomfort as the inappropriate and excess vibration force from whole-body vibration could be transmitted to the head [21]. Approximately 2.4–3.6% of individuals who undergo whole-body vibration approaches develop vertigo [21], muscle soreness [21,55] and discomfort [21,24,56]. Few studies have applied flexi-bar approaches for upper-limb muscle strength enhancement; nonetheless, the vibration frequency and amplitude derived from flexi-bar approaches cannot be used for vibration approaches because flexi-bar vibrations are generated, and their frequencies change based on the participant’s force and skill [57]. In this study, participants were instructed to hold the handheld vibrator in their dominant hand, with their trunk supported on a chair and to maintain their upper limbs in a specific vibration position. Unlike previous studies, our study found that the use of frequency-controlled vibrations from the handheld vibrator could significantly facilitate upper-limb muscle activation without the occurrence of adverse effects such as vertigo and discomfort.

### 4.2. Influence of Vibration Frequency on Upper-Limb Muscle Activation

We found that vibration approaches significantly facilitated upper-limb muscle activation in the vertical and horizontal vibration directions. Unlike no-vibration conditions, vibration conditions enhanced upper-limb muscle activation by 31.4–52.2% and 28.5–54.9% in the vertical and horizontal vibration directions, respectively. Specifically, the maximum muscle activation for FCR was facilitated at 60 Hz in the vertical vibration direction; the maximum muscle activations for FDS, ECR, ECU, biceps, and triceps were induced at 45 Hz in the horizontal vibration direction, respectively. For the deltoid anterior, the maximum muscle activation frequency was 30 Hz in the horizontal direction. Hence, vibration frequencies between 30 and 60 Hz may be used to facilitate maximum muscle activations, and more than half of the muscle groups were activated by high-frequency vibrations (45 and 60 Hz). Recent studies also reported similar findings and showed that a high-frequency vibration has a greater facilitatory effect on muscle activation than a low-frequency vibration in healthy participants and patients with stroke [39,43]. However, the increased muscle activation does not linearly correlate with increasing vibration frequencies [34]. For example, the activation of all muscles at 60 Hz increased the EMGrms by only 0.9–9% rather than doubling it in both vertical and horizontal vibration directions. This phenomenon may be the result of a damping reaction on muscle activation when performing vibration [34]. This corroborates the findings of a recent study, which reported that doubling the acceleration of the applied vibration only facilitates a 3–5% increase in EMGrms [58]. Our findings show that high-frequency vibrations facilitate upper-limb muscle activation, but they may lead to discomfort. For example, whole-body vibration can cause bone frailty, back pain [38,59], resonance injury, and dizziness [21,56]; a vibration frequency of >40 Hz may significantly affect posture control, which results in muscle fatigue [60] and an unstable posture sway [61]. In this study, no participants reported any discomfort or dizziness during or after vibration testing. This may be because the participants were asked to perform all vibration approaches while maintaining the upper limbs in the standard vibration position. Most of the vibration force was thus absorbed by the upper limbs, with little vibration force translating to the body and head. If individuals are unable to hold the vibrator and resist the vibration force during high-frequency vibration approaches, injuries may ensue [62]. Therefore, based on the findings of this study along with those of previous studies, we suggest that a 30-Hz vibration frequency in combination with active muscle participation may be the optimal and safe vibration approach for upper-limb muscle activation. Further studies should be conducted to confirm the benefits of this frequency toward improving muscle strength and function for people with disabilities.

### 4.3. Vibration Direction Affects Upper-Limb Muscle Activation

The FDS and FCR activities at a 45-Hz vibration frequency during horizontal vibrations increased by 9.7% and 6.2%, respectively, compared to that during vertical vibrations. In addition, ECU activation at a 15-Hz vibration frequency was significantly higher in the vertical than in the horizontal vibration direction, by 11.7%. Horizontal vibration has a greater facilitatory effect on FDS muscle activation probably because more muscle activation is required to maintain the handheld vibrator in a fixed position. Additionally, we found that many participants unconsciously flexed their wrists to maintain stability during vibration, which could result in a greater facilitatory effect on FCR activation. Vertical vibration tends to produce a greater facilitatory effect on extensor muscle activation, probably because the device vibrates in the direction of gravity; hence, extensors resist both gravity and the vibration force from the vibrator to maintain the vibration approach posture. Different postures may also result in different vibration directions for the upper limbs during vibration approaches; nonetheless, few studies have investigated the impact of different postures on upper-limb muscle activation. Two recent studies recruited healthy participants to perform vibration approaches in different postures, such as standing on the vibratory platform with arms along the body (squat posture), holding a handrail on top of the vibratory platform in the half-squatting position, and kneeling on the ground with both hands supported over the platform (push-up modified posture) [22,27]. These studies found that muscle strength increases in the push-up modified posture than in the half-squatting posture [27], and the EMGrms for the FDS muscle during squatting was significantly higher than that in the non-vibration condition by 23.3% [22]. However, these findings are difficult to compare with those of the present study because the vibration approaches were different, warranting further investigation.

### 4.4. Study Limitations and Future Perspectives

Our study has some limitations. We developed a frequency-controlled handheld vibrator and indicated the facilitatory effect of vibration approaches on upper-limb muscle activation; however, vibration force transmission and the activation of different upper-limb muscles are complex processes that are influenced by biomechanics [27,38] and vibration protocol parameters (frequency, amplitude, displacement, vibration time, types of vibration, and postures) [42]. Our study only evaluated the effects of vibration frequency and direction on upper-limb muscle activation; hence, the impacts of the other vibration protocol parameters require further investigation. In addition, we suggest that future studies use wearable accelerometers to directly measure the vibration-induced accelerations in target muscle groups of the upper limbs, with analysis of their relationship to muscle activation. For participants, the ratio of male to female participants was not equal in this study. A recent study reported that the body mass of males is higher than that of females and found that vibration-induced accelerations of the body in three dimensions are lower in males [63]. This finding suggests that sex-related differences in body properties and structure may impact the effects of vibration on muscle activation and should thus be considered when applying the vibration approach clinically. Additionally, participants were asked to maintain a firm grip of the vibration device during the experiments. We believe the grip force was approximately constant throughout the experiment, otherwise the participants would have lost grip of the device. However, we did not measure the grip force or pressure underneath, which may have affected the amount of vibration delivered. Future studies could investigate the effects of grip force levels on muscle strength gain due to vibration training. We found many participants unconsciously flexed their wrists to maintain the vibration position of the upper limbs during vibration, which could have been caused by the posture response-induced muscle activation. Future studies should design appropriate vibrators related to the upper limbs, rather than using whole-body vibrators. Furthermore, APDM was applied to validate the vibration frequencies generated by the vibrator between 0 and 60 Hz in this study. However, the maximum sampling frequency for APDM is 128 Hz. To measure vibration frequencies of 60 Hz, a sampling rate that is at least 10 times higher (i.e., ≥600 Hz) is ideal. Theoretically, using a sampling rate that is twice the frequency seems sufficient, but that is the bare minimum. Future studies should use a rate that is at least 10 times larger to collect more accurate measurement of vibration frequency. Meanwhile, due to the COVID-19 pandemic, participant recruitment was limited. Future studies should include a large sample size, along with optimal vibration frequencies and directions during training programs. This would facilitate muscle activation and improve muscle function and functional recovery in frail, older adults, and patients with stroke.

## 5. Conclusions

In this study, we developed a handheld vibrator and validated its positive effects on upper-limb muscle activation. The 45-Hz horizontal vibration approach can induce maximum activations for the FDS, ECR, ECU, biceps, and triceps muscles. Moreover, 60-Hz vertical and 30-Hz horizontal vibrations can facilitate the maximum muscle activations for the FCR and deltoid anterior, respectively. Further studies should be conducted to confirm the benefits of this frequency toward improving muscle strength and function for people with disabilities.

## Figures and Tables

**Figure 1 biology-12-00048-f001:**
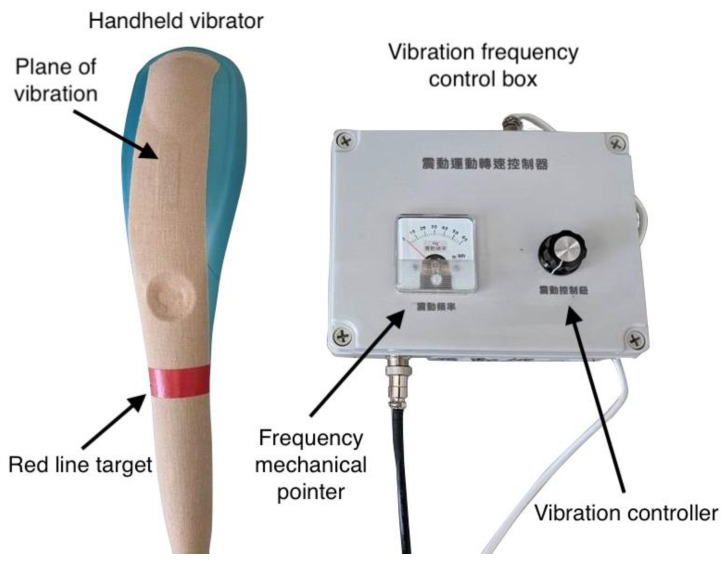
Handheld vibrator and related component parts.

**Figure 2 biology-12-00048-f002:**
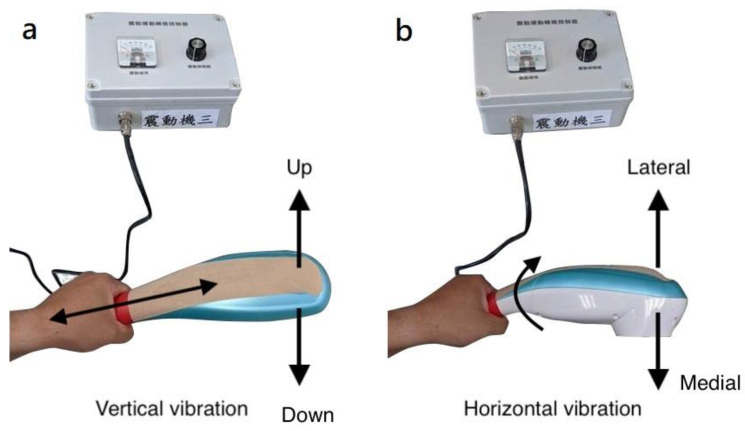
Illustrations for vertical (**a**) and horizontal (**b**) vibration conditions.

**Figure 3 biology-12-00048-f003:**
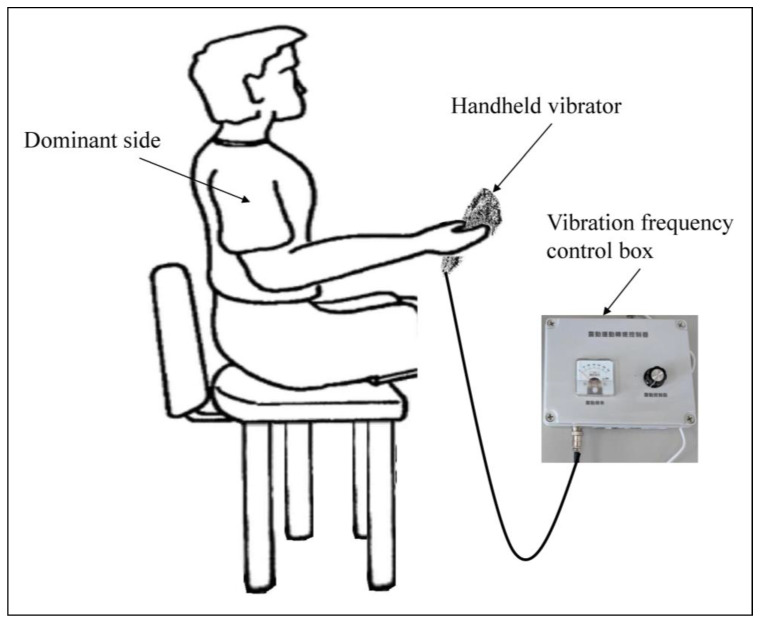
Experimental setting for the vibration.

**Figure 4 biology-12-00048-f004:**
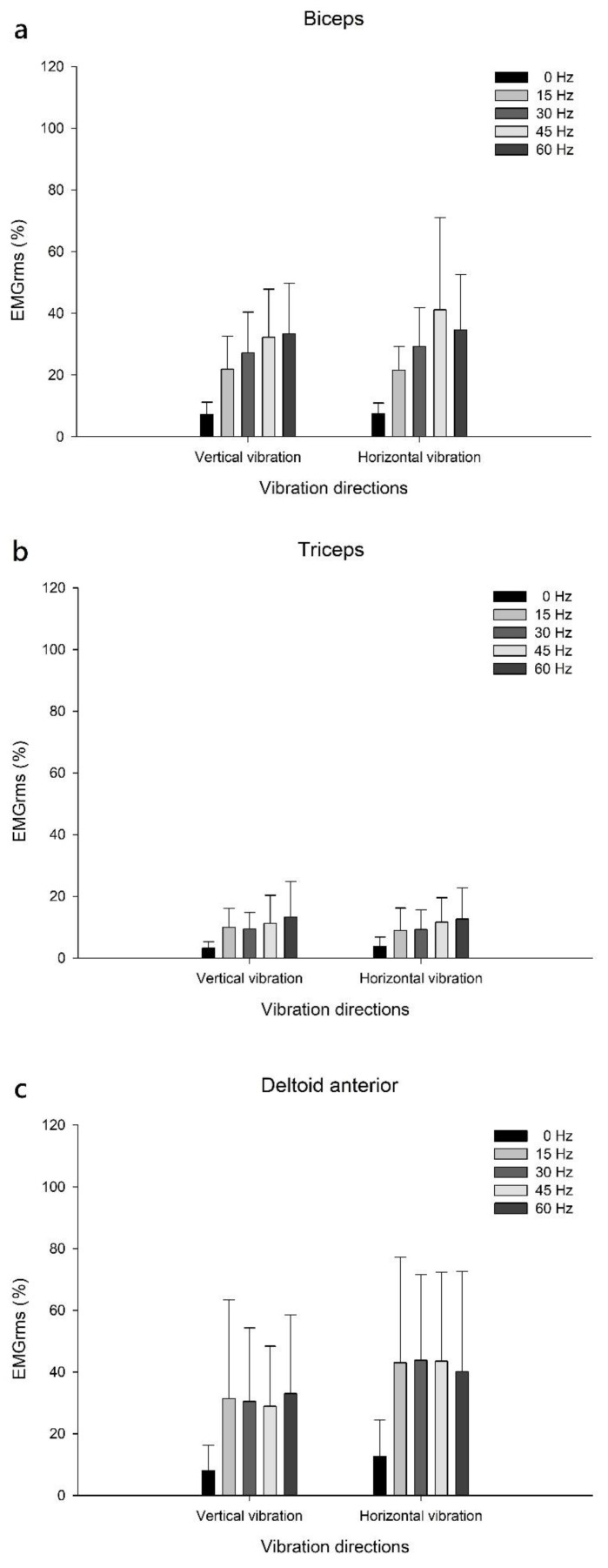
Upper-arm and shoulder muscle activation. Muscle activations in the biceps (**a**), triceps (**b**), and deltoid anterior (**c**) at 0, 15, 30, 45 and 60 Hz vibration frequencies for vertical and horizontal vibration directions. EMGrms, root mean square of surface electromyographic signals.

**Figure 5 biology-12-00048-f005:**
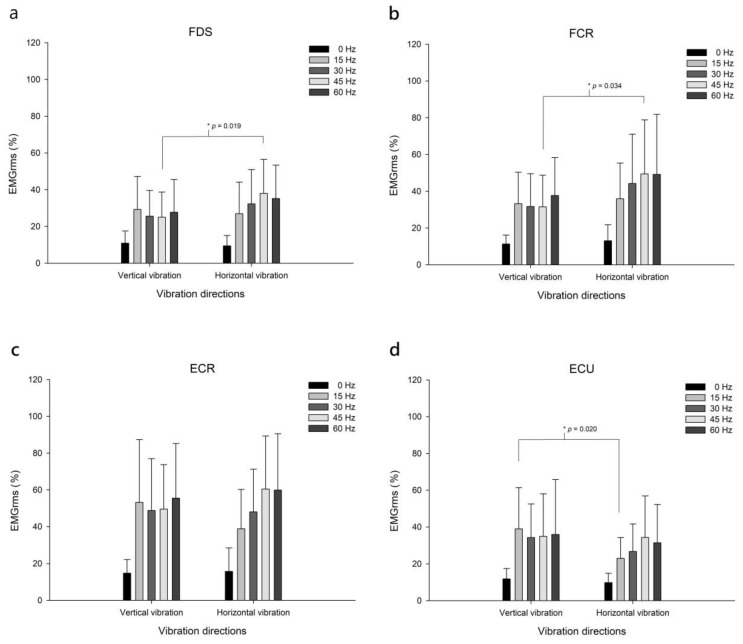
Forearm muscle activation. Muscle activations in the FDS (**a**), FCR (**b**), ECR (**c**), and ECU (**d**) muscles of the forearm at 0, 15, 30, 45, and 60 Hz vibration frequencies in the vertical and horizontal vibration directions. EMGrms, root mean square of surface electromyographic signals. FDS, flexor digitorum superficialis; FCR, flexor carpi radialis; ECR, extensor carpi radialis; ECU, extensor carpi ulnaris.

**Table 1 biology-12-00048-t001:** Demographic characteristics.

	Participants (N = 19)
Age (years)	38.2 ± 14.0
Body weight (kg)	64.3 ± 13.4
Body height (cm)	163.8 ± 9.9
Sex (F/M)	14/5
Dominant side (R/L)	17/2

Data are expressed as mean ± standard deviation or number. Abbreviations: F, female; M, male; R, right; L, left.

**Table 2 biology-12-00048-t002:** Vibration-induced changes in EMGrms (%) compared with those in the no-vibration condition.

	Vertical Vibration ^b^			Horizontal Vibration ^b^			
	Mean (SD)	Wilcoxon Signed-Rank Test ^c^	Friedman Test ^b^	Mean (SD)	Wilcoxon Signed-Rank Test ^c^	Friedman Test ^b^	Mann–Whitney U Test ^a^
		Z-Value (*p* Value)	*X*^2^-Value (*p* Value)		Z-Value (*p* Value)	*X*^2^-Value (*p* Value)	Z-Value (*p* Value)
Flexor digitorum superficialis			35.07 (0.000 ****^b^**)			46.82 (0.000 **^b^)	
0 Hz	10.92 (6.58)	-		9.44 (5.69)	-		158.50 (0.521 ^a^)
15 Hz	29.26 (17.91)	−3.783 (0.000 ****^c^**)		26.95 (17.16)	−3.662 (0.000 ****^c^**)		166.00 (0.672 ^a^)
30 Hz	25.61 (14.00)	−3.743 (0.000 ****^c^**)		32.28 (18.62)	−3.783 (0.000 ****^c^**)		217.00 (0.287 ^a^)
45 Hz	25.01 (13.73)	−3.421 (0.000 ****^c^**)		38.01 (18.54)	−3.823 (0.000 ****^c^**)		261.00 (0.019 *^a^)
60 Hz	27.76 (17.79)	−3.421 (0.000 ****^c^**)		35.17 (18.23)	−3.783 (0.000 ****^c^**)		223.00 (0.215 ^a^)
Flexor carpi radialis			43.41 (0.000 ****^b^**)			34.31 (0.000 ****^b^**)	
0 Hz	11.32 (4.83)	-		13.07 (8.70)	-		188.00 (0.827 ^a^)
15 Hz	33.21 (17.15)	−3.823 (0.000 ****^c^**)		35.92 (19.38)	−3.461 (0.000 ****^c^**)		190.00 (0.782 ^a^)
30 Hz	31.69 (17.81)	−3.783 (0.000 ****^c^**)		44.23 (26.75)	−3.501 (0.000 ****^c^**)		234.00 (0.118 ^a^)
45 Hz	31.52 (17.20)	−3.823 (0.000 ****^c^**)		49.40 (29.40)	−3.622 (0.000 ****^c^**)		253.00 (0.034 *^a^)
60 Hz	37.66 (20.64)	−3.823 (0.000 ****^c^**)		49.20 (32.62)	−3.541 (0.000 ****^c^**)		222.00 (0.226 ^a^)
Extensor carpi radialis			39.70 (0.000 ****^b^**)			59.95 (0.000 ****^b^**)	
0 Hz	14.76 (7.40)	-		15.74 (12.68)	-		175.00 (0.872 ^a^)
15 Hz	53.20 (34.15)	−3.823 (0.000 ****^c^**)		38.83 (21.39)	−3.823 (0.000 ****^c^**)		136.00 (0.194 ^a^)
30 Hz	48.80 (28.18)	−3.783 (0.000 ****^c^**)		48.03 (23.24)	−3.823 (0.000 ****^c^**)		182.00 (0.965 ^a^)
45 Hz	49.5 6 (24.13)	−3.823 (0.000 ****^c^**)		60.47 (28.83)	−3.823 (0.000 ****^c^**)		222.00 (0.226 ^a^)
60 Hz	55.50 (29.73)	−3.823 (0.000 ****^c^**)		59.85 (30.65)	−3.823 (0.000 ****^c^**)		194.00 (0.693 ^a^)
Extensor carpi ulnaris			38.82 (0.000 **^b^)			49.72 (0.000 **^b^)	
0 Hz	11.83 (5.64)	-		9.83 (5.05)	-		145.50 (0.307 ^a^)
15 Hz	38.97 (22.46)	−3.823 (0.000 ****^c^**)		22.95 (11.22)	−3.823 (0.000 ****^c^**)		101.00 (0.020 *^a^)
30 Hz	34.21 (18.26)	−3.823 (0.000 ****^c^**)		26.64 (14.94)	−3.823 (0.000****^c^**)		133.00 (0.166 ^a^)
45 Hz	34.91 (23.07)	−3.823 (0.000 ****^c^**)		34.25 (22.56)	−3.823(0.000****^c^**)		183.00 (0.942 ^a^)
60 Hz	35.90 (29.90)	−3.702 (0.000 ****^c^**)		31.39 (20.81)	−3.823 (0.000****^c^**)		173.00 (0.827 ^a^)
Biceps			54.61 (0.000 ****^b^**)			53.72 (0.000 **^b^)	
0 Hz	7.22 (3.92)	-		7.37 (3.57)	-		191.50 (0.748 ^a^)
15 Hz	21.89 (10.69)	−3.823 (0.000 ****^c^**)		21.56 (7.64)	−3.823(0.000 ****^c^**)		185.00 (0.895 ^a^)
30 Hz	27.12 (13.23)	−3.823 (0.000 ****^c^**)		29.24 (12.58)	−3.823(0.000 ****^c^**)		202.00 (0.530 ^a^)
45 Hz	32.21 (15.63)	−3.823 (0.000 ****^c^**)		41.18 (29.81)	−3.823(0.000 ****^c^**)		202.00 (0.530 ^a^)
60 Hz	33.38 (16.27)	−3.823 (0.000 ****^c^**)		34.64 (17.89)	−3.823(0.000 ****^c^**)		189.00 (0.804 ^a^)
Triceps			35.74 (0.000 **^b^)			48.46 (0.000 **^b^)	
0 Hz	3.19 (2.02)	-		3.74 (3.00)	-		191.50 (0.748 ^a^)
15 Hz	9.94 (6.11)	−3.823 (0.000 ****^c^**)		8.89 (7.30)	−3.823 (0.000 ****^c^**)		157.00 (0.493 ^a^)
30 Hz	9.35 (5.39)	−3.743 (0.000 ****^c^**)		9.27 (6.34)	−3.823 (0.000 ****^c^**)		168.00 (0.715 ^a^)
45 Hz	11.17 (9.14)	−3.662 (0.000 ****^c^**)		11.64 (7.88)	−3.823 (0.000 ****^c^**)		192.00 (0.737 ^a^)
60 Hz	13.27 (11.54)	−3.823 (0.000 ****^c^**)		12.62 (10.05)	−3.823 (0.000 ****^c^**)		176.00 (0.895 ^a^)
Deltoid anterior			39.11 (0.000 ****^b^**)			43.91 (0.000 ****^b^**)	
0 Hz	8.01 (8.14)	-		12.72 (11.66)	-		246.50 (0.054 ^a^)
15 Hz	31.36 (32.06)	−3.823 (0.000 ****^c^**)		43.04 (34.16)	−3.823 (0.000 ****^c^**)		241.00 (0.077 ^a^)
30 Hz	30.46 (23.82)	−3.823 (0.000 ****^c^**)		43.79 (27.79)	−3.823 (0.000 ****^c^**)		240.00 (0.082 ^a^)
45 Hz	28.91 (19.48)	−3.823 (0.000 ****^c^**)		43.55 (28.83)	−3.823 (0.000 ****^c^**)		237.00 (0.099 ^a^)
60 Hz	32.99 (25.55)	−3.823 (0.000 ****^c^**)		40.16 (32.46)	−3.702 (0.000 ****^c^**)		205.00 (0.474 ^a^)

* Significant difference *p* < 0.05. ** Significant difference *p* < 0.01. ^a^ Mann–Whitney U test was performed; ^b^ Friedman test was performed; ^c^ Wilcoxon signed-rank test was performed; significantly greater than that in the no-vibration (0 Hz) condition. Vibrations were applied at frequencies of 15, 30, 45, and 60 Hz in the vertical and horizontal directions. EMGrms, root mean square of surface electromyographic signals; SD, standard deviation.

## Data Availability

The datasets are available from the corresponding author upon reasonable request.

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
