# Peer review of "Analysis of Vibration Frequency and Direction for Facilitating Upper-Limb Muscle Activity"

_biology, 2022, doi:10.3390/biology12010048_

Round 1

Reviewer 1 Report

Overall comments:

The paper titled “Investigation of Vibration Frequency and Direction for Facilitating Upper-Limb Muscle Activity” presents an investigation of varying different parameters of focal vibration as applied on the muscle. The paper measured EMG for different vibration frequencies, muscle locations, and directions, for 19 adults. The novel contribution is the investigation the effects of two different directions of vibration on the muscle as well.

Overall, the paper is well-written, and the results and discussion have been presented well. There are a few details which need to be clarified. These are minor changes. Therefore, I recommend this article for publication with minor revisions.

Specific comments:

Figure 2. Illustration of vertical versus horizontal vibration applied is not clear. Please revise it. Please show the device resting on a muscle with the arrows showing whether they are perpendicular or parallel to the surface/skin/muscle.

Line 133. Please mention the sampling frequency used for the accelerometers for the APDM sensor used to benchmark the frequency. I believe their maximum is 128 Hz. For measuring frequency of 60 Hz, ideally one should use a sampling rate of 10X of that, i.e. 600 Hz or more. Theoretically, using 2X the frequency seems sufficient but that is the bare minimum, and practically, one should use at least 10X. Otherwise, the 60 Hz vibration measured at 128 Hz sampling rate could be off by as much as 19%! It is fine if this study has used a lower frequency, but it is an important detail that must be mentioned.

Line 136-137. Please clarify what the pointer is showing. Is it showing realtime frequency measurements of vibration from the sensor mounted on the vibration device? Is it simply showing voltage applied to the vibration motor, correlated with frequency measurements previously?

Line 153. Please add one or two sentences on how the vibration device was applied to the muscles in terms of pressure. Was the device loosely held over the muscle? Was the device gently or firmly pressed against the muscle? Did the authors ensure that consistent pressure was applied each time? If not, then this should be a consideration for future work. The amount of pressure applied affects the quantity of vibration delivered, so it is important that the authors highlight whether, and how, this was considered in their study.

Please add this study limitation if relevant. If the accelerometer sampling rate was indeed less than 600 Hz, the authors should mention that more accurate measurement of frequency would be obtained in the future by using accelerometers with a higher sampling rate (approximately 10X the expected frequency).

Please add this study limitation if relevant. If the authors did not take care to ensure that the vibration device was applied with consistent pressure, then in a future study, care must be taken to apply vibration with consistent pressure for each scenario for each subject. Perhaps they can use a rig or strap that applies pressure in a repeatable fashion.

Author Response

Reviewer #1:

The paper titled “Investigation of Vibration Frequency and Direction for Facilitating Upper-Limb Muscle Activity” presents an investigation of varying different parameters of focal vibration as applied on the muscle. The paper measured EMG for different vibration frequencies, muscle locations, and directions, for 19 adults. The novel contribution is the investigation the effects of two different directions of vibration on the muscle as well.

Overall, the paper is well-written, and the results and discussion have been presented well. There are a few details which need to be clarified. These are minor changes. Therefore, I recommend this article for publication with minor revisions.

General response: We appreciate your recommendation regarding the publication of our paper in Biology.

Specific comments:

Figure 2. Illustration of vertical versus horizontal vibration applied is not clear. Please revise it. Please show the device resting on a muscle with the arrows showing whether they are perpendicular or parallel to the surface/skin/muscle.

Response: Thank you for your suggestion. We have revised Figure 2 to make it clear. We have also revised the Methods section as follows: “Participants used their dominant hand to hold the vibrator at the red line. Participants held the vibrator in two different ways (Fig. 2a and 2b) so that the direction of vibration was either up-down or lateral-medial”.

Line 133. Please mention the sampling frequency used for the accelerometers for the APDM sensor used to benchmark the frequency. I believe their maximum is 128 Hz. For measuring frequency of 60 Hz, ideally one should use a sampling rate of 10X of that, i.e. 600 Hz or more. Theoretically, using 2X the frequency seems sufficient but that is the bare minimum, and practically, one should use at least 10X. Otherwise, the 60 Hz vibration measured at 128 Hz sampling rate could be off by as much as 19%! It is fine if this study has used a lower frequency, but it is an important detail that must be mentioned.

Response: Thank you for your comment. The sampling frequency for the APDM was 128 Hz. We have added this detail in the Discussion section.

Line 136-137. Please clarify what the pointer is showing. Is it showing realtime frequency measurements of vibration from the sensor mounted on the vibration device? Is it simply showing voltage applied to the vibration motor, correlated with frequency measurements previously?

Response: The pointer is showing the voltage applied to the vibration motor, which is correlated with the frequency measurement via APDM.

Line 153. Please add one or two sentences on how the vibration device was applied to the muscles in terms of pressure. Was the device loosely held over the muscle? Was the device gently or firmly pressed against the muscle? Did the authors ensure that consistent pressure was applied each time? If not, then this should be a consideration for future work. The amount of pressure applied affects the quantity of vibration delivered, so it is important that the authors highlight whether, and how, this was considered in their study.

Response: Thank you for your comments. Participants held the vibration device during the experiment. The participant held the vibrator in two different ways (Fig. 2a and 2b) so that the direction of vibration was either up-down or lateral-medial. Participants were asked to maintain a firm grip of the vibration device during the experiments. We believe the level of grip force was approximately constant throughout the experiment, otherwise the participants would have lost grip of the device; however, we did not measure the grip force or pressure underneath. We have added this information in the limitation section of the Discussion and suggest that future studies could investigate this issue.

Please add this study limitation if relevant. If the accelerometer sampling rate was indeed less than 600 Hz, the authors should mention that more accurate measurement of frequency would be obtained in the future by using accelerometers with a higher sampling rate (approximately 10X the expected frequency).

Response: Thank you for your comment. We have added this limitation regarding the accelerometer sampling rate being less than 600 Hz, and suggest that more accurate measurement of frequency can be obtained by using accelerometers with a higher sampling rate in the last paragraph of the Discussion.

Please add this study limitation if relevant. If the authors did not take care to ensure that the vibration device was applied with consistent pressure, then in a future study, care must be taken to apply vibration with consistent pressure for each scenario for each subject. Perhaps they can use a rig or strap that applies pressure in a repeatable fashion.

Response: Thank you for your suggestion. We have added this limitation in the Discussion and suggest that future studies could investigate this issue.

Reviewer 2 Report

Every article we want to publish is based on some investigation. Consider losing the word ‘investigation’ from your title, or replacing it with ‘analysis’. But this is just an irrelevant suggestion. Feel free to neglect.

Regarding ‘aged-induced frailty’. Can you discuss, and cite relevant literature, how much the frailty is really influenced by age as opposed to lifestyle?

Please, expand your introduction with a detailed explanation on how the vibration approach is implemented on patients, include graphics.

Due to pandemic you recruited 19, instead of 34, participants. How did it influence the parameters, such as the effect size, alpha, power?

It would contribute to the professionally of your paper, if you figured out how to remove the backgrounds from the photos in Figs. 1 and 2.

In row 153-156, you describe the participants’ position. Can you add a simple graphic of a figurine (stick-figure) showing the position?

Avoid starting sentences with acronyms. 

Thank you for providing the results in both the graphs and tables. Many studies avoid plotting graphs and resort to tables only. Good job overall, and looking forward to reading more of your papers in the future.

Author Response

Reviewer #2:

Dear authors,

Every article we want to publish is based on some investigation. Consider losing the word ‘investigation’ from your title, or replacing it with ‘analysis’. But this is just an irrelevant suggestion. Feel free to neglect.

Response: Thank you for the comment. We have adjusted the title to “Analysis of Vibration Frequency and Direction for Facilitating Upper-Limb Muscle Activity”.

Regarding ‘aged-induced frailty’. Can you discuss, and cite relevant literature, how much the frailty is really influenced by age as opposed to lifestyle?

Response: Based on our clinical experiences and literature review, we found that age and lifestyle (sedentary behaviors) are both associated with frailty. However, most studies investigated and revealed age-induced physiologic changes in vicious loops, leading to poor muscle strength and muscle atrophy (sarcopenia). This results in muscle weakness, low physical activity, and functional impairments, and increases dependency levels in daily living for the elderly. All this information has been added in the Introduction section.

Please, expand your introduction with a detailed explanation on how the vibration approach is implemented on patients, include graphics.

Response: We have update the Introduction section with a detailed description on how to implement the vibration approach in patients from previous studies.

Due to pandemic you recruited 19, instead of 34, participants. How did it influence the parameters, such as the effect size, alpha, power?

Response: Thank you for your question. We recalculated the sample size for a Wilcoxon test in G*Power, which showed a total sample size of 7 and power of 0.83 after entering the mean (SD) of group 1 [10.92 (6.58)] and group 2 [29.26 (17.91)]. Based on this, we believe the 19 participants is sufficient.  

It would contribute to the professionally of your paper, if you figured out how to remove the backgrounds from the photos in Figs. 1 and 2.

Response: Thank you for your suggestion. We have removed the backgrounds from the photos in Figs. 1 and 2.

In row 153-156, you describe the participants’ position. Can you add a simple graphic of a figurine (stick-figure) showing the position?

Response: As requested, we have added a simple graphic as Figure 3 to show the participants’ position (Methods section).

Avoid starting sentences with acronyms. 

Response: We have reviewed whole paper, and added the word “The” before “FDS and FCR activities” in the Discussion section. We have also asked the Editage team to help us to check all starting sentences and avoid acronyms at the start of sentences.

Thank you for providing the results in both the graphs and tables. Many studies avoid plotting graphs and resort to tables only. Good job overall, and looking forward to reading more of your papers in the future.

Response: Thank you for your comment. In 2023, we hope to use the findings of this study to investigate the benefits of vibration on enhancing muscle strength and functionality for frail older adults and stroke patients. We believe our findings will help clinical staff to improve the health of older individuals with frailty and those who have experienced stroke.

Reviewer 3 Report

Review

Investigation of Vibration Frequency and Direction for Facilitating 

Upper-Limb Muscle Activity

Biology 2072030

The article aims to present the effect of vibration frequency and direction on upper limb muscle activation using a portable vibrator.

The article includes: introduction, research material and methods (participants, research device and data processing, experimental procedures and positioning of participants, EMG analysis, statistical analyses), results, discussion, conclusions and bibliography.

The objective of the study is mentioned from the beginning, it being that of presenting the positive effects that the use of a portable vibrator can have on the muscle strength of the elderly, at the level of the upper limbs.

Regarding the research methodology, it considers a sample of 19 people aged 38.2 ± 14.0 years (14 women and 5 men).

Of the 58 bibliographic references, 95% appeared after the year 2000, 16 of them - in the last five years.

The novelty of this article is the verification of the impact only at the level of the upper limbs, because, based on the study of the specialized literature, the authors found that exercises with vibrations for the whole body (vibration platform for the whole body) also produce negative effects (vertigo, discomfort).

The results of the study are thoroughly substantiated. The authors point out that, unlike experiments in which vibrations were induced throughout the body and which generated a series of negative effects, in the present experiment no such effects were identified.

The conclusions highlight the positive effects of using such a portable vibrator on the activation of the muscles of the upper limbs.

Strong points:

Several types of tests are used/applied;

The groups of muscles on which we intervened with vibrations and the results obtained for each are presented;

It is the first study on the activation of different upper limb muscles induced by different directions of vibration;

No adverse effects were reported;

It is a starting point for further research/opens up/provides opportunities for future research

The study is conducted in accordance with the Declaration of Helsinki.

Weaknesses:

Reduced sample

I agree with the publication of the article, which I find interesting, thoroughly documented and detailed, with the mention that these assessments come from a specialist in physical education and sports. In this sense, I consider the views/opinions of some specialists in the field of medicine or biology to be much more grounded.

Author Response

Review

Investigation of Vibration Frequency and Direction for Facilitating 

Upper-Limb Muscle Activity

Biology 2072030

The article aims to present the effect of vibration frequency and direction on upper limb muscle activation using a portable vibrator.

The article includes: introduction, research material and methods (participants, research device and data processing, experimental procedures and positioning of participants, EMG analysis, statistical analyses), results, discussion, conclusions and bibliography.

The objective of the study is mentioned from the beginning, it being that of presenting the positive effects that the use of a portable vibrator can have on the muscle strength of the elderly, at the level of the upper limbs.

Regarding the research methodology, it considers a sample of 19 people aged 38.2 ± 14.0 years (14 women and 5 men).

Of the 58 bibliographic references, 95% appeared after the year 2000, 16 of them - in the last five years.

The novelty of this article is the verification of the impact only at the level of the upper limbs, because, based on the study of the specialized literature, the authors found that exercises with vibrations for the whole body (vibration platform for the whole body) also produce negative effects (vertigo, discomfort).

The results of the study are thoroughly substantiated. The authors point out that, unlike experiments in which vibrations were induced throughout the body and which generated a series of negative effects, in the present experiment no such effects were identified.

The conclusions highlight the positive effects of using such a portable vibrator on the activation of the muscles of the upper limbs.

Strong points:

Several types of tests are used/applied;

The groups of muscles on which we intervened with vibrations and the results obtained for each are presented;

It is the first study on the activation of different upper limb muscles induced by different directions of vibration;

No adverse effects were reported;

It is a starting point for further research/opens up/provides opportunities for future research

The study is conducted in accordance with the Declaration of Helsinki.

Weaknesses:

Reduced sample

I agree with the publication of the article, which I find interesting, thoroughly documented and detailed, with the mention that these assessments come from a specialist in physical education and sports. In this sense, I consider the views/opinions of some specialists in the field of medicine or biology to be much more grounded.

Response: Thank you for your positive feedback and recommendation. Regarding the reduced sample size, we recalculated the sample size for a Wilcoxon test using G*Power which showed a total sample size of 7 and power of 0.83 after entering the mean (SD) value of group 1 [10.92 (6.58)] and group 2 [29.26 (17.91)]. Therefore, the power of 0.83 for the 19 recruited participants seemed enough in this study. In 2023, we hope to use the findings of this study to investigate the benefits of vibration on enhancing muscle strength and functionality for frail older adults and stroke patients. We believe our findings will help clinical staff to improve the health of older individuals with frailty and those who have experienced stroke.

Round 2

Reviewer 3 Report

I believe that the revised form of the article meets the criteria to be published. Good luck in your further research.